# Pathophysiological Implications of Imbalances in Fibroblast Growth Factor 23 in the Development of Diabetes

**DOI:** 10.3390/jcm10122583

**Published:** 2021-06-11

**Authors:** Javier Donate-Correa, Ernesto Martín-Núñez, Ainhoa González-Luis, Carla M. Ferri, Desirée Luis-Rodríguez, Víctor G. Tagua, Carmen Mora-Fernández, Juan F. Navarro-González

**Affiliations:** 1Unidad de Investigación, Hospital Universitario Nuestra Señora de Candelaria, 38010 Santa Cruz de Tenerife, Spain; jdonatecorrea@gmail.com (J.D.-C.); emarnu87@gmail.com (E.M.-N.); ainhoa.gonaluz@gmail.com (A.G.-L.); carlamferri@gmail.com (C.M.F.); vgarciat@ull.edu.es (V.G.T.); carmenmora.fdez@gmail.com (C.M.-F.); 2GEENDIAB (Grupo Español para el Estudio de la Nefropatía Diabética), Sociedad Española de Nefrología, 39008 Santander, Spain; 3Escuela de Doctorado y Estudios de Posgrado, Universidad de La Laguna, 38200 San Cristóbal de La Laguna, Spain; 4Servicio de Nefrología, Hospital Universitario Nuestra Señora de Candelaria, 38010 Santa Cruz de Tenerife, Spain; desireeluis@gmail.com; 5Instituto de Tecnologías Biomédicas, Universidad de La Laguna, 38010 San Cristóbal de La Laguna, Spain; 6REDINREN (Red de Investigación Renal-RD16/0009/0022), Instituto de Salud Carlos III, 28029 Madrid, Spain

**Keywords:** diabetes, chronic kidney disease, fibroblast growth factor 23, inflammation, pancreatic ß cell, immune cells, αKlotho

## Abstract

Observational studies have associated the increase in fibroblast growth factor (FGF) 23 levels, the main regulator of phosphate levels, with the onset of diabetes. These studies open the debate on the plausible existence of undescribed diabetogenic mechanisms derived from chronic supraphysiological levels of FGF23, a prevalent condition in chronic kidney disease (CKD) and end-stage renal disease (ESRD) patients. These maladaptive and diabetogenic responses to FGF23 may occur at different levels, including a direct effect on the pancreatic ß cells, and an indirect effect derived from the stimulation of the synthesis of pro-inflammatory factors. Both mechanisms could be mediated by the binding of FGF23 to noncanonical receptor complexes with the subsequent overactivation of signaling pathways that leads to harmful effects. The canonical binding of FGF23 to the receptor complex formed by the receptor FGFR1c and the coreceptor αKlotho activates Ras/MAPK/ERK signaling. However, supraphysiological concentrations of FGF23 favor non-αKlotho-dependent binding of this molecule to other FGFRs, which could generate an undesired overactivation of the PLCγ/CN/NFAT pathway, as observed in cardiomyocytes and hepatocytes. Moreover, the decrease in αKlotho expression may constitute a contributing factor to the appearance of these effects by promoting the nonspecific activation of the PLCγ/CN/NFAT to the detriment of the αKlotho-dependent Ras/MAPK/ERK pathway. The description of these mechanisms would allow the development of new therapeutic targets susceptible to be modified by dietary changes or by pharmacological intervention.

## 1. Introduction

Diabetes mellitus (DM) is a major health concern for the world population. Worldwide, about 351.7 million people on working age (20–64 years) have diagnosed or undiagnosed DM disease, a number that is expected to rise to 417.3 million by 2035 and 486.1 by 2045 [1]. The sedentary lifestyle, consumption of processed food, and rate of childhood obesity are contributing factors to this prevalence of diabetes. Type 2 diabetes mellitus (T2DM) is the most common type of diabetes, accounting for around 90% of all diabetes. This disease covers a wide pathophysiological spectrum, including insulin resistance (IR) and the appearance of alterations in pancreatic ß-cell function. This cell, responsible for the synthesis of insulin, shows alterations in its functionality from very early stages in the progression of the disease, even before the diagnosis of T2DM. As IR progresses, the ability of the ß cell to maintain euglycemia decreases due to a progressive loss of its ability to synthesize and/or secrete insulin. Despite important advances in recent decades, our understanding of the pathophysiology of T2DM remains incomplete. The approaches currently proposed for the prevention and treatment of this disease involve avoiding or reducing risk factors associated with the loss of ß-cell functionality.

The incidence of T2DM results from a combination of genetic, environmental, and behavioral risk-factors that include sedentary lifestyle and diet. Related to diet, small elevations in the levels of Pi in blood also constitute a risk factor for the appearance of prediabetes situations, particularly, impaired glucose tolerance and IR, as well as for the development of T2DM [2]. Similarly, elevated Pi and calcium x Pi product in animal models are also related with the appearance of IR [3]. 

Moreover, the levels of Pi are also associated with a pro-inflammatory status, being related with inflammatory markers such as C-reactive protein (CRP) and interleukin (IL) 6 [4]. These molecules, together with the proinflammatory cytokines tumor necrosis factor α (TNFα) and IL1ß, as well as acute-phase reactants, are involved in the promotion of IR and also in ß-cell function impairment and apoptosis in T2DM [5,6].

The above-mentioned Pi serum range is maintained by diverse hormones that regulate the intestinal uptake, its mobilization from bone, and the renal excretion. Importantly, the pathophysiological repercussions of Pi imbalances also involve to these regulatory factors [7,8,9,10]. Disbalances in phosphataemic regulatory-factors traditionally related to an increase in morbidity are the decrease in calcitriol (the active form of vitamin D) and the increase in parathyroid hormone (PTH) levels. However, the description of the fibroblast growth factor 23 (FGF23) as a new regulatory component has changed our vision of the regulation of mineral metabolism [11]. Currently, this hormone is considered the main regulator of phosphorus and vitamin D metabolism [12]. FGF23 is secreted from bones, especially by osteoblasts and osteocytes, after phosphate intake and acts primarily on the kidneys to inhibit phosphate reabsorption in urine. FGF23 also inhibits calcitriol renal synthesis and the secretion of PTH in the parathyroid glands [12]. Only the full-length intact form of FGF23 (iFGF23), a 32-kDa glycoprotein consisting of 227 amino acids, is biologically active. However, circulating FGF23 can be also found as inactive forms which result from a proteolytic cleavage between Arg179 and Ser180 during secretion, generating non-active N- and C-terminal fragments [13].

Phosphate-rich diets have been shown to increase circulatory levels of FGF23 in both humans and rodents [14,15]. Similarly, high extracellular Pi enhances FGF23 production in osteoblastic cell lines [16]. Circulating FGF23 level is regulated at transcriptional and translational level and also by posttranslational modification. Therefore, O-linked glycosylation of the threonine 178 position by N-acetylgalactosaminyltransferase 3 (GalNAc-T3) encoded by *GALNT3* inhibits the cleavage of FGF23 protein and contributes to an increase in serum full-length FGF23 level. A mechanism of phosphate sensing and regulation of FGF23 production have recently been proposed pointing to the *GALNT3* as a phosphate-responsive gene that protects FGF23 from proteolytic degradation in response to increments in Pi levels [17].

Beyond its role as a phosphataemic regulator, circulating FGF23 is a strong predictor of disease progression and mortality in patients with chronic kidney disease (CKD), in whom higher levels of this hormone are prevalent, with an emerging role in cardiovascular disease [18,19,20]. However, FGF23 has also been associated with obesity, dyslipidemia, visceral adiposity, IR markers, and an increased risk of metabolic syndrome both in childhood and in adult populations [21,22,23]. Moreover, observational studies show higher circulatory FGF23 levels in patients with DM [24,25,26,27,28,29,30,31]. These clinical studies have been conducted in CKD patients [24,25], but also in individuals with preserved renal function [21,26]. Although these findings are merely associative and do not demonstrate causality or directionality, they allow us to open the debate on the existence of a determining role of imbalances in FGF23 levels on the incidence of DM or its complications [32], with direct actions on the endocrine system.

## 2. Clinical Studies Relating FGF23 to Insulin Resistance and Diabetes Mellitus

Patients with early stages of glucose alterations (IR and prediabetes) present higher FGF23 levels compared to normal glucose tolerant subjects [33]. Furthermore, some works have found higher circulating FGF23 levels in patients with DM or in normoglycemic subjects with a history of DM in first-degree relatives [26,27,28,29,30,31,34] (Table 1). On the other side, a few clinical studies with small sample sizes have failed to find an association between the presence of diabetes and circulating FGF23 levels [35].

It is important to note that these works have used different assay kits that are commercially available to measure circulating FGF23; some of them detect iFGF23 alone, and others detect the C-terminus of the protein that is present both in the iFGF23 and in the C-terminal fragment (collectively named cFGF23) [36]. This must be taken into account in comparative studies.

Wahl et al. [30] carried out a multicenter cross-sectional study involving 3756 participants with mild to moderate CKD (20–70 mL/min/1.73 m^2^) and reported that patients with DM (the type of diabetes was not considered) had higher plasma levels of cFGF23 (172.4 (114.3–277.2) vs. 121.9 (84.0–198.8) RU/mL, *p* < 0.001) and also experienced an earlier onset of FGF23 excess (≥100 RU/mL). Diabetes was associated with FGF23 plasma levels even after adjusting for demographic, clinical, and laboratory data. This association was maintained in the group of subjects with an eGFR > 60 mL/min/1.73 m^2^ (99.2 (74.2–131.5) vs. 84.4 (61.7–111.9) RU/mL, *p* < 0.05) [30].

Mirza et al. [21], in a cross-sectional study including two community-based cohorts (964 and 946 subjects) of elderly whites, investigated the relationships between serum iFGF23 levels and different markers of obesity and insulin utilization. Authors found that higher FGF23 levels were associated with obesity, dyslipidemia (lower HDL cholesterol and apolipoprotein A1, higher triglyceride), and increased risk of presenting metabolic syndrome. Moreover, a 1-standard deviation increase in log FGF23 was associated with 8–12% increases in insulin levels and homeostatic model assessment of insulin resistance (HOMA-IR) index in both cohorts. Although these associations were not statistically significant in multivariable adjusted models, the results suggested for the first time that hyperinsulinemia and IR might be contributing factors to the increase in circulating FGF23 levels in the diabetes population.

Garland et al. [24] carried out a cross-sectional study to determine the associations between cFGF23, IR, and coronary artery calcification in 72 CKD patients in stages 3–5 not receiving insulin therapy. Patients with values of HOMA-IR > 2.2 presented greater cFGF23 (179.7 vs. 109.6 RU/mL, *p* = 0.03) without changes in Pi levels. They also observed that log cFGF23 levels were significantly higher as the number of metabolic syndrome components increased (*p* = 0.03), being 30% higher in patients with 4–5 components of metabolic syndrome (2.31 ± 0.54 vs. 2.01 ± 0.45 log RU/mL; *p* = 0.02). Moreover, cFGF23 was positively correlated with HOMA-IR (r = 0.25, *p* = 0.04). Multivariable linear regression adjusted for calcitriol, kidney function (eGFR, urinary albumin to creatinine ratio (UACR)), and PTH revealed that IR was a risk factor for greater log cFGF-23 levels (log HOMA-IR β = 0.37, 95% CI: 0.14–0.59, *p* = 0.002).

Fernandez-Real et al. [37] also reported a positive correlation between the levels of cFGF23 and HOMA-IR levels (r = 0.35, *p* = 0.006) in 314 non-CKD obese subjects (BMI > 30 kg/m^2^). This correlation was absent for iFGF23. In a multiple linear regression analysis, body mass index (BMI) (*p* = 0.001) contributed independently to 45% of the variance in circulating cFGF23 levels in these subjects. Similarly, in another cross-sectional study by Gutierrez et al. [38] including 1261 community-dwelling adults, those participants in the highest category of BMI had 9.5 RU/mL higher cFGF23 levels than those in the lowest.

Gateva et al. [33] designed a cross-sectional study to evaluate the differences in calcium–phosphate metabolism markers in a small group of obese patients with prediabetes (*n* = 39) compared to an age- and BMI-matched normoglycemic control group (*n* = 41) with no differences in serum creatinine levels and eGFR. They found significantly higher levels of iFGF23 in the first group (10.4 ± 10.7 vs. 5.8 ± 7.3 pg/mL, *p* = 0.03) and a weak positive correlation to fasting blood glucose (r = 0.224; *p* = 0.048). Moreover, they also found higher levels of iFGF23 in those patients with IR, with an almost statistical significance (9.5 ± 10.1 vs. 5.2 ± 7.3 pg/mL, *p* = 0.05).

Hanks et al. [26] determined the associations of cFGF23 with markers of insulin utilization (resistin, adiponectin, HOMA-IR) and inflammation (interleukin (IL) 6, IL10, and high sensitivity-CRP (hsCRP)) and anthropometrics in a cross-sectional study including a large cohort of 1040 community-dwelling adults. Authors found a positive association of cFGF23 levels with HOMA-IR and with indices of obesity (BMI and waist circumference) and with all the inflammatory markers considered in individuals without CKD, but not among individuals with CKD. They also found a higher incidence of DM in higher tertiles of circulating cFGF23 levels (*p* < 0.001). Hanks et al. [28] concluded that elevated FGF23 concentrations could serve as a subclinical marker of metabolic perturbations (diabetes, dyslipidemia, and obesity) in individuals with normal kidney function.

Winther et al. [34] investigated serum levels of iFGF23 and phosphate during a euglycemic–hyperinsulinemic clamp in a small study population that included 30 subjects equally distributed in three groups: lean, glucose-tolerant obese healthy subjects, and patients with overt T2DM. Baseline serum iFGF23 concentrations did not differ significantly between the subgroups. However, after insulin infusion, serum iFGF23 concentration significantly increased only in the diabetic group (4.7 ± 4.5 pg/mL, *p* = 0.009). This increase correlated with the increase in insulin levels (r = 0.83, *p* = 0.003). This small but significant increase in serum iFGF23 induction by hyperinsulinism in patients with T2DM under euglycemic conditions points to the contribution of insulin to FGF23 production.

**Table 1 jcm-10-02583-t001:** Main clinical studies showing associations of FGF23 level markers of IR and DM.

Population	FGF23 Analyte	Type of Study	Main Finding	Ref.
Two independent cohorts of elderlies (964 and 946 subjects)	iFGF23	Cross-sectional	FGF23 was associated with markers of obesity, MS, insulin levels, and HOMA-IR index.	[21]
68 adolescents with simply obesity	iFGF23	Cross-sectional	FGF23 negatively correlated with HOMA-IR and fasting insulin level.	[22]
72 patients with CKD stages 3–5	cFGF23	Cross-sectional	FGF23 positively correlated with HOMA-IR and with the number of MS components. IR constituted a risk factor for greater log cFGF23 levels.	[24]
1040 community-dwelling adults	cFGF23	Cross-sectional community-based	Diabetes was prevalent in higher tertiles of FGF23. FGF23 correlated with HOMA-IR and markers of obesity and inflammation in subjects with preserved renal function.	[26]
604 patients with CKD stages 2–4	cFGF23	Cross-sectional	Positive association between the presence of diabetes and serum FGF23 levels.	[27]
780 healthy older men (>60 years)	cFGF23	Cross-sectional	Elevated FGF23 was associated with diabetes.	[28]
1719 normoglycemic participants, 312 with a first degree FHD	iFGF23	Cross-sectional	Subjects with first-degree FHD presented higher levels of FGF23 accompanied by increases in serum insulin levels and HOMA-IR values.	[29]
3756 patients with mild to moderate CKD	cFGF23	Multicenter cross-sectional	Patients with diabetes had increased FGF23 levels and presented an earlier onset of FGF23 excess.	[30]
133 patients with CVD	iFGF23 and cFGF23	Cross-sectional	Diabetes was more prevalent in the higher tertiles of both iFGF23 and cFGF23 determinations.	[31]
39 prediabetes obese patients and 41 age- and BMI-matched normoglycemic group	iFGF23	Case/control	FGF23 was higher in patients with prediabetes and IR.	[33]
10 obese individuals with T2DM, 10 glucose-tolerant obese healthy individuals, and 10 lean subjects.	iFGF23	Case/control	After euglycemic–hyperinsulinemic clamp, only subjects with DM presented a significant increase in serum FGF23 levels, which correlated with insulin variation.	[34]
314 non-CKD obese subjects	cFGF23	Cross-sectional	FGF23 and HOMA-IR levels were positively correlated.	[37]

FGF23, fibroblast growth factor 23; MS, metabolic syndrome; HOMA-IR, homeostatic model assessment of insulin resistance; IR, insulin resistance; CKD, chronic kidney disease; FHD, family history of diabetes; DM, diabetes mellitus; CVD, cardiovascular disease.

In contrast to the aforementioned studies, Wojcik et al. [22] found negative correlations between circulating FGF23 and HOMA-IR in non-insulin-resistant subjects. Authors determined the correlation between iFGF23 serum levels and body composition, blood pressure and selected parameters of glucose, and insulin and fat metabolism in a group of 68 adolescents (mean age 13.9 years) with simple obesity. In this study, authors found negative correlations between circulating FGF23 and fasting insulin level, and HOMA-IR (r = −0.3 and r = −0.29, respectively; *p* < 0.05 for both). Consistent with this work, a study published in 2018 by Bär et al. [39] revealed a clear negative correlation of plasma insulin with FGF23 in a group of healthy pregnant women without manifest IR or hyperinsulinemia. Moreover, authors provided strong evidence supporting the suppressive effect of insulin on FGF23 synthesis in vitro as well as in mice and humans. To explain the discrepancy with previous studies, where insulin-resistant individuals had higher FGF23 serum levels, authors suggested that intact insulin signaling is required for this suppressive effect by insulin [23]. Along with the IR status, other cofounding factors that influence the synthesis of FGF23 coexist in patients with DM. These include inflammation, preexisting CKD or coronary heart disease, obesity, and high leptin levels, and all of them may have a further impact on FGF23 secretion [18,19,20,21,25,26]. Other studies not designed ad hoc for this purpose also showed the existence of an association of FGF23 levels with diabetes. Hu et al. [29] carried out a study designed to assess the value of iFGF23 for identifying subclinical atherosclerosis in normoglycemic individuals with a first-degree family history of diabetes (FHD). The study included 1719 subjects, 312 of them with a first-degree FHD, all of them with normal kidney function and without impaired glucose regulation or diabetes. Serum iFGF23 level were much higher in subjects with a first-degree FHD than in those without a FHD (29.2 (25–35.99) vs. 28.30 (22.9–35) pg/mL, *p* = 0.006). The increase in iFGF23 levels in these subjects was accompanied by increases in serum insulin levels and HOMA-IR values. Similarly, Schoppet et al. [28], in a cross-sectional analysis of the STRAMBO cohort designed to assess the association of cFGF23 levels with mineral metabolism parameters and abdominal aortic calcification (AAC) in men, also reported an association of diabetes with elevated cFGF23 levels in a group of 780 healthy older men (>60 years) (*p* < 0.005). Levels of cFGF23 according to the presence of DM were not reported. Similarly, Donate-Correa et al. [33], in a study designed to detect differences in FGF23 levels according to the presence of atherosclerosis or vascular calcification in 133 patients with cardiovascular disease (mean eGFR: 78.8 ± 24 mL/min/1.73 m^2^), reported a higher prevalence of patients with diabetes in the higher tertiles of circulating FGF23 levels, both for iFGF23 and cFGF23 determinations (*p* < 0.01, for both). Vervloet et al. [27], in a study designed to determine the association of FGF23 with demographic and clinical parameters using multivariable regression models, also found an association of cFGF23 levels with the presence of diabetes in a group of 604 patients with moderate to severe kidney disease (ß = 0.159 RU/mL, *p* = 0.035).

## 3. FGF23 Signaling Pathway

FGF23 belongs to the FGF superfamily, which in humans consists of 22 signaling peptides that participate in a broad diversity of biological processes. FGF23, together with FGF19 and FGF21, form the particular group of endocrine (hormone-like) FGFs segregated from the wide FGF ligand superfamily by phylogenetic and sequence analysis [40,41]. Since both FGF19 and FGF21 are implied in the regulation of lipid and glucose metabolism [42,43,44,45,46], it is plausible that FGF23 may also be involved in some metabolic processes, especially in the metabolism of glucose.

The lack of active heparan–sulfate (HS) binding domains in endocrine FGFs prevents the formation of hydrogen bindings with the HS-rich extracellular matrix [47] and allows entry to the bloodstream. However, this feature of endocrine FGFs also determines a low affinity for their cell surface tyrosine kinase receptors, termed FGF receptors (FGFRs). There are four FGFRs (FGFR1–4) that present a similar structure and a high degree of amino acid sequence homology [48] and that are practically ubiquitous, being expressed in multiple organs and tissues [49]. A new FGFR called FGFR5 has recently been added to this group of receptors, which lacks the tyrosine-kinase domain and which is believed to regulate FGFR1 responses [50]. The alternative splicing of the codifying genes produces several FGFR subtypes, including b and c subtypes of FGFR1 to FGFR3 [51]. To enhance the affinity for FGFRs in their target organs, endocrine FGFs use Klotho proteins as cofactors. Since FGFRs are expressed in a wide range of tissues, tissue-specific expression of Klotho proteins is considered to be the determinant for an organ to be targeted by endocrine FGFs. Furthermore, it has been reported that the soluble form of αKlotho, generated by proteolytic cleavage of the membrane-anchored form, may also function as a coreceptor for FGFR1c [49,52], although the significance of soluble αKlotho in the transmission of FGF23 signaling is unknown.

The components of the canonical receptor complex for FGF23 are FGFR1c and the membrane-anchored protein αKlotho, which is expressed in several restricted tissues, including the kidneys and the parathyroid glands [53]. Recently, the crystal structure of the FGFR1c/αKlotho complex has been described, demonstrating that αKlotho is a nonenzymatic molecular scaffold for FGF23 signaling [52]. In addition to kidneys and parathyroid glands, the expression of αKlotho has also been detected in the choroid plexus, vascular tissue, peripheral blood cells (PBCs), and recently in pancreatic ß cells [53,54,55,56,57,58,59]. The presence of αKlotho in ß cells suggests that this protein may play a role related to the synthesis and/or release of insulin through its role as a coreceptor for FGF23. Furthermore, it is also plausible that circulating FGF23 directly mediates off-target effects independently of αKlotho, and it has been proposed that, at high concentrations, FGF23 is capable of establishing independent low-affinity αKlotho binding to FGFRs other than FGFR1c, thus causing deleterious effects on multiple organs and tissues.

FGF23 regulates circulating Pi levels by decreasing blood Pi and calcitriol levels [13,60,61]. In the kidneys, FGF23 reduces reabsorption of phosphate from the urine by reducing the abundance of type IIa and IIc sodium–phosphate cotransporters (NaPis) in the apical membrane of epithelial cells in the proximal renal tubule [12,61]. Additionally, FGF23 also reduces renal calcitriol synthesis by reducing the transcription of renal 1α-hydroxylase (CYP27B1), the key enzyme for 1.25 (OH)_2_D_3_ synthesis [13,60,61]. These actions are mediated by FGFRs that can activate several intracellular signal transduction pathways, including the extracellular signal-regulated kinase (ERK), protein kinase B (Akt) and phospholipase C-γ (PLCγ) pathways [62]. The canonical αKlotho-dependent signal transduction that FGF23 is believed to employ to regulate phosphate and vitamin D metabolism is the Ras/MAPK/ERK [49]. The transduction is initiated by activation through autophosphorylation of tyrosine kinase enzymes in the cytoplasmic tail of FGFR1c, which induces the activation of the Ras/MAPK/ERK pathway and, posteriorly, the expression of the early growth response 1 (EGR1) protein, which acts as a differential transcription factor.

## 4. Potential Diabetogenic Actions of FGF23

Most of the studies that relate FGF23 to the appearance of imbalances in glucose and insulin metabolism are merely descriptive, and currently, there are no mechanisms explaining these relationships. One of the possible explanations may be the combined effects of high levels of FGF23 and reduced expression of its specific cofactor αKlotho.

In conditions characterized by the presence of supraphysiological levels of FGF23, the unspecific binding of this hormone could mediate off-target effects in tissues and organs not previously considered to be targets, explaining part of these effects. Several data support the existence of this complementary mechanism of action of FGF23 that can be activated in certain circumstances. A few alternatives to canonical FGF23 signal transduction have been proposed. αKlotho can be found in blood and cerebrospinal fluid as a soluble protein, and as mentioned above, it has been proposed that this soluble form may act as a widely available cofactor for the FGFR1c receptor [52]. A second possibility is that the new onset or the stimulation of the expression of αKlotho in tissues where it is not generated or is in a very low proportion could generate a functional receptor complex when it is colocalized with the ubiquitously expressed FGFR1c. Finally, signal transduction in response to FGF23 may occur independently of αKlotho by binding to receptors FGFR2, 3, and 4 since only FGFR1c requires the presence of αKlotho to bind FGF23 with sufficient affinity [63]. This possibility has been demonstrated in a small group of cells and tissues including myocardial tissue, hepatocytes, and neutrophils, which only express FGFR2 and FGFR4 [64,65,66].

The existence of this multiplicity of bindings for FGF23, even in the same target organ if it co-expresses different FGFRs, has important repercussions on the potential activation of different intracellular signals and, consequently, on the effects elicited by this hormone. The PLCγ/calcineurin (CN)/nuclear factor of activated T cells (NFAT) signaling pathway is a noncanonical pathway activated by non-αKlotho-dependent FGF23 binding. The overactivation of this pathway results in hypertrophic effects in cardiac myocytes that, at the clinical level, is associated with the appearance of left ventricular hypertrophy (LVH), and in the induction of an inflammatory response in hepatocytes [18,64,65]. Similarly, the suppression of PTH expression in the parathyroid glands, which is canonically mediated by the binding of FGF23 to the FGFR1c/αKlotho receptor complex, can also occur through the activation of the PLCγ/CN/NFAT pathway independently of αKlotho [67]. However, the chronic activation of this pathway derived from an excess of FGF23 generates maladaptive effects leading to hyperplasia of parathyroid cells and an increase in the secretion of PTH [68]. To date, the potential diabetogenic effects of the activation of this pathway by noncanonical binding of FGF23 remain unexplored. Although these effects can be at very different levels, there are two possibilities that we consider to deserve attention.

### 4.1. Effects on Pancreatic ß Cell

FGF signaling plays an important role in the maintenance of ß-cell physiology and glucose homeostasis, with disorders in the intracellular signaling response being associated with the onset and progression of diabetes [69,70]. FGF1 and 2 specifically stimulate insulin secretion in rodent pancreatic ß cells [71,72]. Conversely, the impairment of FGFR1 signaling leads to the appearance of the diabetic phenotype in mice, indicating that this receptor, but not FGFR2, has a crucial role in the control of glucose homeostasis [73].

The binding of FGF23 to the pancreatic ß cell would allow the existence of a mechanism of cellular regulation which is not yet known. This possibility has been reinforced by demonstration of the expression of coreceptor αKlotho, together with FGFR1c and FGFR2b, in the pancreatic ß cell [18,73]. Moreover, the recently identified FGFR5 is also expressed in adult pancreas [74,75] and exerts modulatory effects on FGFR1 activity, enhancing tyrosine phosphorylation of the Ras/MAPK/ERK signaling pathway [76].

Thus, FGF23 could act in the ß cell through this canonical signaling pathway and exert modulatory actions on the production and/or release of insulin in this cell. Furthermore, the feedback could be closed by an inhibition of the production of FGF23 by insulin: both insulin and insulin-like growth factor 1 (IGF1) are capable of suppressing FGF23 production through activation of phosphatidyl inositol triphosphate kinase (PI3K)/Akt/forkhead box protein O1 (FOXO1) signaling [39].

The PLCγ/CN/NFAT pathway is an important regulator of multiple biological functions, including the regulation of ß-cell growth and function as well as the biosynthesis and secretion of insulin. Administration of the CN inhibitors CsA and FK506 to rodents or humans induces hyperglycemia and hypoinsulinemia derived from a reduction in insulin biosynthesis and secretion [77]. However, sustained activation of the PLCγ/CN/NFAT pathway provokes similar deleterious effects on ß-cell proliferation, growth, and function [78]. Therefore, the activation of noncanonical pathways by FGF23-intracellular-signal transmission under certain conditions in which FGF23 is overexpressed and/or in which αKlotho, the membrane and/or the systemic soluble form, is absent or poorly expressed could generate a harmful effect by causing an overactivation of the CN pathway, resulting in a malfunction of the pancreatic ß cell. In this sense, αKlotho has been recently shown to play a role in glucose metabolism. Transgenic, αKlotho-deficient mice exhibit pancreatic islet atrophy with reduced pancreatic insulin mRNA and protein levels and serum insulin concentrations [79]. By contrast, αKlotho overexpression in mice resulted in increased plasma membrane retention of the Ca^2+^-permeable, transient receptor potential cation channel V2 (TRPV2), enhanced calcium entry and glucose-induced intracellular calcium response, and insulin secretion, whereas knockdown of αKlotho attenuated these effects [80].

### 4.2. Effects on Inflammation

Chronic low-grade inflammation and activation of the innate immune system constitute key factors in the pathogenesis of diabetes mellitus [5,81,82]. Diverse inflammatory parameters are elevated in diabetic patients and constitute strong predictors of the development of this disease [83,84,85]. Given the prevalence of inflammation in T2DM and its role in the development of the disease, the potential effects of increased levels of FGF23 on the immune system may be of clinical relevance.

Similar to Pi, elevated FGF23 plasma levels have been independently associated with higher levels of inflammatory markers in patients with CKD or other inflammatory diseases. Results of the Chronic Renal Insufficiency Cohort (CRIC) showed that higher FGF23 levels are associated with higher levels of the inflammatory markers CRP, IL6, TNFα, and fibrinogen and with a higher odds ratio for severe inflammation independent of mineral metabolism and renal function [25]. This association is not limited to CKD, and the levels of FGF23 have been positively correlated with TNFα [80] and with IL6 [21] in general populations, and with CRP levels in the elderly [21,26]. Furthermore, inflammation is a major trigger of FGF23 production [86,87,88,89], indicating the potential existence of a feedback mechanism between FGF23 and inflammatory markers.

In human PBCs, which express αKlotho and the receptors FGFR1c, 2, and 4, FGF23 is able to inhibit calcitriol production, a well-known immune system modulator, through the activation of the Ras/MAPK/ERK signaling pathway [90]. Although this inhibition mediated by FGF23 may contribute to the suppression of innate immunity, indirectly explaining the effects of FGF23 on the inflammatory status, experimental data indicate that FGF23 can directly promote the synthesis of inflammatory factors. Although the effects of FGF23 in the immune system are unknown, mice models of FGF23 excess have revealed an overactivation of genes regulating inflammation such as *TGFß1*, *TNF*, *IL1ß*, and *NF-kB* [91]. Experimental studies show that hepatocytes treated with FGF23 increase CRP and IL6 expression, and that this effect is blocked by anti-FGFR4 or Cyclosporin A (CsA), pointing to the activation of the PLCγ/CN/NFAT signaling pathway as a mechanism responsible for these actions [64]. Similarly, the injection of the carboxi-tail peptide of FGF23, which prevents iFGF23 signaling, in a diabetic nephropathy mouse model reduced the renal expression as well as the serum levels of inflammatory cytokines IL6 and TNFα without affecting serum FGF23 and Pi levels [92]. These taken together, there is evidence of a role of FGF23 in the modulation of the inflammatory response, although the regulatory mechanism as well as its physiological function are barely understood.

The activation of noncanonical pathways could be involved in the association between FGF23 and inflammation. The pro-inflammatory effect promoted by FGF23 in hepatocytes described above is mediated by an αKlotho-independent activation of FGFR4 [64]. On the other hand, FGF23 has been reported to affect macrophages and stimulate TNFα expression via the Ras/MAPK/ERK signaling pathway, also through an αKlotho-independent mechanism [93]. The macrophage cell line RAW264.7 in nonpolarized M0 and anti-inflammatory M2 stages expresses FGFR1c and low levels of αKlotho. However, pro-inflammatory M1 macrophages, polarized with LPS and IFNγ, upregulated αKlotho gene expression and produced TNFα through the activation of ERK1/2 following exposure to FGF23 [93]. This finding suggests that high serum FGF23 levels might amplify the inflammatory response induced by primed macrophages, but to date, the contribution of FGF23 to a pro-inflammatory status derived from the activation of the PLCγ/CN/NFAT pathway in macrophages, and also other immune cells such as lymphocytes, is unknown. αKlotho has been previously detected in CD4+ lymphocytes, and marked reductions in its expression have been related to aging [57], but currently, there are no data on the involvement in the function of either murine or human T lymphocytes.

## 5. Conclusions

Higher levels of FGF23, a frequent situation in CKD and HD patients, are strongly and independently associated with a higher risk of morbidity and mortality in renal patients, but also in individuals with preserved kidney function. Moreover, observational studies have associated this increment with the appearance of IR markers and the incidence of diabetes. This article aimed to review these works and, at the same time, propose a mechanistic hypothesis that could explain these observational studies in which experimental support is scarce. Although there is currently no mechanism to explain all these associations, the existence of a regulatory effect of hormone FGF23 on the pancreatic ß cell and/or on the induction of inflammation cannot be ruled out. In this way, the elevation of its concentration to supraphysiological levels, motivated by pathological states, would generate a maladaptive effect in the ß cell and immune cells that could translate into a decrease in insulin production, or even in the loss of viability of this cell type. Undoubtedly, all these ideas must eventually be contrasted through more observational and, above all, experimental studies.

Moreover, many of the clinical studies have been limited by a focus on kidney disease populations and/or by a small sample size. Most of them also lack determinations of αKlotho levels, which should be considered for a proper interpretation of the effects elicited by FGF23. The few studies that, together with FGF23, also determine the levels of this protein show that serum αKlotho levels are reduced in patients with prediabetes, although without reaching statistical significance. It has been described that αKlotho itself is implicated in the development of IR, and its gene polymorphism seems to be an important contributor to the MS origin [94,95]. Although αKlotho levels have been associated with IR [80,96,97], the impact of this protein on insulin sensitivity seems not to be direct [98], suggesting that this regulation is possibly via FGF23.

Despite these findings, many aspects need to be clarified. For instance, high levels of FGF23 are present in some forms of rare disorders known as hypophosphatemic rickets (HR), which have different etiopathogenesis, including genetic diseases, and share a similar phenotype caused by excessive renal phosphate loss. The forms of HR caused by the excess of FGF23 are the X-linked form (XLH) caused by an inactivating mutation in the PHEX gene, tumor-induced osteomalacia (TIO), epidermal nevus syndromes, and fibrous dysplasia in McCune–Albright syndrome [99]. None of these diseases is associated with an increase in the incidence of diabetes. This discrepancy is likely due to the absence of concomitant contributing factors which are absent in these phenotypes, including hyperphosphatemia and a state of deficiency in αKlotho. These disarrangements are probably necessary for the appearance of imbalances in glucose and insulin metabolism linked to the excess of FGF23.

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
