# Peer review of "Pathophysiological Implications of Imbalances in Fibroblast Growth Factor 23 in the Development of Diabetes"

_jcm, 2021, doi:10.3390/jcm10122583_

Round 1

Reviewer 1 Report

This review deals, according to the title, with the role of FGF23 in type2 diabetes. In fact, most of the review is devoted to the biology and various associations of FGF23 with other conditions that also to some extent associate with T2D.

Some of the concepts presented are outdated such as the signaling of FGF23 to the proximal tubule, there is extensive work from O Moe and R Erben on direct signaling of FGF23 and the expression of klotho in this segment.

The part on FGF23 and T2D is thin and mostly speculative. Not going much further than some hypotheses and associations. There is no substantial data presented that make a causative role of FGF23 in T2D plausible. Thus, the authors should either provide such evidence or consider to restructure the review/change title and discuss the potential roles of FGF23 in a different setting.

The review lacks major literature form the last 5 years.

Author Response

Thank you very much for your comments. According to them, we have made several changes that we hope will increase the quality of it. Specifically, we have discussed in greater depth the clinical trials that relate FGF23 levels to alterations in glucose metabolism and insulin utilization, as well as the development of diabetes. We have also updated the bibliography including the works by O. Moe and R. Erben related to FGF23 signaling.

Reviewer 2 Report

This review sums up the effects of FGF23 and αKlotho on various organs. The authors briefly describe the different pathways by which FGF23/Klotho act. They present some hypotheses regarding potential causal links between FGF23 and diabetes occurence or complications. Their analysis of the data from the literature could be a little more detailed. For example the authors do not consider the differences in FGF23 levels between patients with X-linked hypophosphatemia or various severity of renal insufficiency. Similarly they could mention that while αKlotho expression is markedly decreased in renal insufficiency it is normal in X-linked hypophosphatemia.

Author Response

Thank you very much for your valuable comments. According them, we have made several changes that we hope will increase the quality of the manuscript. Specifically, we have discussed in greater depth the clinical studies that relate FGF23 levels to alterations in glucose metabolism and insulin utilization, as well as the development of diabetes. We have also considered the presence of CKD as an important complication related to the increase in FGF23 levels and its plausible contributing effects on the relations between this hormone and the appearance of diabetes or its complications. We also mention X-linked hypophosphatemia, which is a condition not associated with a higher incidence of diabetes despite being characterized by high levels of FGF23, and we discuss the plausible explanation for this lack of association.

Round 2

Reviewer 1 Report

The authors discuss that higher FGF23 levels have been found in patients with type 2 diabetes with and without CKD and conclude that there is a correlation and that its indicates that FGF23 may have a determining effect on the incidence of DM. Both statements are wrong and the authors apparently do not understand the simple nature of an association. All findings cited are simple associations, they do not demonstrate any causality and no directionality. The confusion of correlations and associations goes through the whole manuscript.

The authors contiune from there claiming that many studies have shown that FGF23 contributes to the development of metabolic syndrome and DM. Again, genetic diseases of high FGF23 are not characterized by metabolic syndrome of DM and again, these studies are simple association studies.

In the next paragraph the association of FGF23 with the onset of DM is discussed. The authors should discuss here that insulin suppresses FGF23 and that this may be attenuated in states of insulin resistance. Moreover, obesity and high leptin levels, a common finding in pats with DM, drives FGF23 levels. Likewise, inflammation increases FGF23 as discussed later.

Overall, the case against FGF23 as a driver of DM is weak and the authors should probably be more cautious in their conclusions.

Author Response

Dear reviewer,

Again, thank you very much for your comments. According to your suggestions, we have made several changes in the manuscript. 

1- The authors discuss that higher FGF23 levels have been found in patients with type 2 diabetes with and without CKD and conclude that there is a correlation and that its indicates that FGF23 may have a determining effect on the incidence of DM. Both statements are wrong and the authors apparently do not understand the simple nature of an association. All findings cited are simple associations, they do not demonstrate any causality and no directionality. The confusion of correlations and associations goes through the whole manuscript.

We have replaced the term correlation by association when appropriate and clarified that the studies carried out do not allow to conclude a causal effect on the incidence of DM derived from the elevation of FGF23.

2- The authors continue from there claiming that many studies have shown that FGF23 contributes to the development of metabolic syndrome and DM. Again, genetic diseases of high FGF23 are not characterized by metabolic syndrome of DM and again, these studies are simple association studies.

As the reviewer points up, high FGF23 levels are present in a group of rare disorders with different etiopathogenesis, including genetic diseases, and with a similar phenotype caused by excessive renal phosphate wasting and known as hypophosphatemic rickets (HR). The forms of HR caused by the excess of FGF23 are the X-linked form (XLH) caused by an inactivating mutation in the PHEX gene, tumour-induced osteomalacia (TIO), epidermal nevus syndromes and fibrous dysplasia in McCune–Albright syndrome Neither of these diseases is associated with an increase in the incidence of diabetes. In the manuscript, we argue that this discrepancy is likely due to the absence of concomitant contributing factors which are absent in these phenotypes, including hyperphosphatemia and a state of deficiency in αKlotho. These disarrangements are probably necessary for the appearing of imbalances in glucose and insulin metabolism linked to the excess of FGF23.

3- In the next paragraph the association of FGF23 with the onset of DM is discussed. The authors should discuss here that insulin suppresses FGF23 and that this may be attenuated in states of insulin resistance. Moreover, obesity and high leptin levels, a common finding in pats with DM, drives FGF23 levels. Likewise, inflammation increases FGF23 as discussed later.

We have included a short paragraph that analyzes the suppressive effect of insulin on the synthesis of FGF23. We also comment on the possibility that this suppression may be influenced by the existence of a state of IR. We also point to other cofounding factors in the patient with DM that may influence the synthesis of FGF23: inflammation, preexisting CKD or coronary heart disease, obesity and high leptin levels.

4- Overall, the case against FGF23 as a driver of DM is weak and the authors should probably be more cautious in their conclusions.

This article aims to review the published works that deal with the association of the increase in FGF23 and the appearance of alterations in the use of insulin or diabetes. At the same time, we propose a mechanistic hypothesis that could explain these observational studies in which experimental support is scarce. Undoubtedly, these studies do not allow to conclude a causal effect on the incidence of DM derived from the elevation of FGF23 and all the mechanistic hypothesis exposed must eventually be contrasted through more observational and, above all, experimental studies. This statement is now in the Conclusions section.